# Neighborhood Versions of Geometric–Arithmetic and Atom Bond Connectivity Indices of Some Popular Graphs and Their Properties

**Muhammad Shafii Abubakar [1]**, **Kazeem Olalekan Aremu [1,2,\*]** and **Maggie Aphane [1]**

1   Department of Mathematics and Applied Mathematics, Sefako Makgatho Health Sciences University, Ga-Rankuwa, P.O. Box 60, Pretoria 0204, South Africa

2   Department of Mathematics, Usmanu Danfodiyo University, Sokoto P.M.B. 2346, Nigeria

\*   Correspondence: aremukazeemolalekan@gmail.com or aremu.kazeem@udusok.edu.ng

**Abstract:** In this article, we introduce the neighborhood versions of two classical topological indices, namely neighborhood geometric–arithmetic and neighborhood atom bond connectivity indices. We study the graph-theoretic properties of these new topological indices for some known graphs, e.g., complete graph $K_n$, regular graph $R_n$, cycle graph $C_n$, star graph $S_n$, pendant graph, and irregular graph and further establish their respective bounds. We note that the neighbourhood geometric–arithmetic index of $K_n$, $R_n$, $C_n$, and $S_n$ is equal to the number of edges. The neighborhood atom bond connectivity index of an arbitrary simple graph $\mathcal{G}$ is strictly less than the number of edges. Our results contribute to the literature in this direction.

**Keywords:** degree-based topological indices; neigborhood degree sum; neigborhood topological indices; geometric–arithmetic index; atom bond connectivity index

## 1. Introduction

Let $\mathcal{G}$ be an arbitrary graph, a topological index is a function $f : \mathcal{G} \rightarrow \mathbb{R}$. Topological indices (TI) are graph invariants and are employed to describe the topology of graphs. Topological indices are very important in mathematical chemistry because they are used to model physiochemical properties of molecules and compounds. The molecules are illustrated as a simple connected graph; with atoms of chemical compound denoting vertices and the chemical bonds between them as edges. In 1947, Harold Wiener [1] introduced the first TI related to molecular branching, Wiener [1] showed that his TI is closely related to the boiling points of alkane molecules, his QSPR and QSAR analysis showed that it is also related with other quantities such as the parameters of its critical point, the density, surface tension, viscosity of its liquid phase, and van der Waals surface area of the molecule. The Wiener index is defined as

$$\mathcal{W} = \frac{1}{2} \sum_{i=1}^{n} \sum_{j=1}^{n} (d_{ij}), \tag{1}$$

where $d_{ij}$ represents off-diagonal elements of $d$. The success of Wiener's work influenced the study of other TI of chemical graphs such as the Randic index which was introduced by Milan Randic [2] in 1975. The Randic index is also called the connectivity index is given as

$$R(\mathcal{G}) = \sum_{uv \in E(\mathcal{G})} \frac{1}{\sqrt{d_u d_v}} \tag{2}$$

where $d_u$ and $d_v$ are the degrees of vertices of the graph $\mathcal{G}$. The Randic index $R(\mathcal{G})$ was used to determine the topology of linear and branched alkanes with eight or less carbon atoms. Randic [2] further stated that the degree of branching of the molecular skeleton is very important in determining some molecular parameter such as boiling points of hydrocarbons. Computations from the $R(\mathcal{G})$ showed that the numerical parameter obtained is in satisfactory agreement with the kovats retention index whose purpose is to determine the retention time interpolated between adjacent *n*-alkanes. This was a major success because it showed the efficiency of the $R(\mathcal{G})$ in its applications to physical properties of molecules and compounds (see [2–5] for more details). For this reason, numerous researchers have devoted attention to the development of several TI (for example, Zagreb index [6–8], second Zagreb index [8], forgotten index [9], harmonic index [10], GA index [11], and ABC index [12]) and their applications to model molecules of higher carbon atoms. In 2014, Gutman et al. [13] studied the reciprocal of $R(G)$ in (2). They defined the reciprocal form as follows:

$$RR(\mathcal{G}) = \sum_{uv \in E(\mathcal{G})} \sqrt{d_u d_v} \tag{3}$$

and applied it to physico-chemical properties of octane isomers such as the standard enthalpy of formation and boiling points. Although $RR(\mathcal{G})$ has less computational advantage to $R(\mathcal{G})$. Furthermore, Vukičević and Furtula [11] introduced a new type of degree based TI known as geometric–arithmetic $GA(\mathcal{G})$ index which is given by

$$GA(\mathcal{G}) = \sum_{uv \in E(\mathcal{G})} \frac{2\sqrt{d_u d_v}}{d_u + d_v}. \tag{4}$$

They discovered that the $GA$ index can be used as a predictive tool in quantity structure–property relationship (QSPR) and quantity structure–activity relationship (QSAR) analysis. The predictive power was further tested on some physicochemical properties of octanes and the results obtained showed that the $GA$ index gives a better predictive power than the $R(\mathcal{G})$ in (2) (see [14–21] for details). Estrada et al. [12] introduced an important TI (which seems to be advantageous in applications) known as the Atom Bond Connectivity index (*ABC* index for short). The *ABC* index is defined as:

$$ABC(\mathcal{G}) = \sum_{uv \in E(\mathcal{G})} \sqrt{\frac{d_u + d_v - 2}{d_u d_v}}. \tag{5}$$

It was noted in [12] that unlike the $R(\mathcal{G})$, the *ABC* index does not show the level of branching of the molecule; rather, it described the heats of formation of alkanes, which gave a good correlation coefficient in the QSPR model ($r = 0.997$) (see [12,17,22–26] for more details).

Recently, Mondal et al. [27] developed the concept of neighborhood TI. The neighborhood TI summed the degree of a distinct vertex over a distinct neighbor set. This new approach seems to be better in application especially for determining degeneration of TI (see [27]). In the pioneer work on neigborhood topological indices by Mondal et al. [27], they studied the neighborhood versions of forgotten index, Zagreb index, second Zagreb index, and hyper Zagreb index. They further applied it to model the regression analysis of *n*-octane isomers. Since then, neighborhood version of other classical TI have been studied (reciprocal Randic index, sum connectivity index, redefined third Zagreb index, Randic index, and symmetric division degree index. See [28] for more details).

Despite the rapid development of neighborhood versions of TI, we note that there are some classical TI that their neighborhood versions are yet to be studied. Motivated by this fact and the works of Vukičević et al. [11], Estrada et al. [12] and Mondal et al. [27], we introduce the neighborhood geometric–arithmetic index and atom bond connectivity index. We study the graph theoretic properties of the new TI and establish their respective bounds. Our result compliment other results in the literature.

## 2. Preliminaries

A *graph* $\mathcal{G}$ can be defined as a triple $(V, E, f)$ where $V$ is a finite nonempty set called set of vertices, $E$ is a finite set (may be empty) called set of edges and $f$ is the incidence function that assigns to each edge $e \in E$ a one-element subset $\{v\}$ or two element subset $\{u, v\}$. The *degree* of a vertex $v$ on a graph $\mathcal{G}$, denoted by $d_v$ is the total number of edges associated with $v$. Let $N(v)$ denotes the set of *neighbors* of $v$, the sum of the degrees of the neighbors of $v$ is called the *neighbor degree sum*, denoted by $\mathcal{D}_v$ and given by

$$\mathcal{D}_v = \sum_{u \in N(v)} \mathcal{D}_u. \tag{6}$$

The *order* of a graph is the number of vertices contained in the graph while the *size* of the graph is the number of edges contained in the graph, thus $|V(\mathcal{G})| = n$ and $|E(\mathcal{G})| = m$. A *simple* graph $\mathcal{G}$ is a graph that contains no multiple edges or loops. A *connected* graph is a graph that has a path between every pair of vertices in the graph. A *regular* graph $R_n$ is a simple graph where every vertex has the same vertex degree. A *complete* graph $K_n$ is a simple graph in which every pair of distinct vertices is connected by a unique edge. A *cycle* graph $C_n$ is a simple graph that consists of a single cycle. A *pendant vertex* is a vertex that has degree 1 i.e., $d_v = 1$. An *irregular graph* is a graph in which all vertices have a unique degree. A *star graph* $S_n$ is a connected graph in which there exists at most one vertex with a degree greater than one [29,30]. Henceforth, all graphs considered in this article are simple and finite.

**Definition 1.** *Let $\mathcal{G}$ be a graph, $\mathcal{D}_u$ and $\mathcal{D}_v$ be the neighbor degrees of vertices u and v, respectively. The*

*(i) neighborhood first Zagreb index [27] is given by*

$$M_1^*(\mathcal{G}) = \sum_{u \in V(\mathcal{G})} [\mathcal{D}_u + \mathcal{D}_v], \tag{7}$$

*(ii) neighborhood hyper Zagreb index [27] is given by*

$$M_H^*(\mathcal{G}) = \sum_{u,v \in E(\mathcal{G})} [\mathcal{D}_u + \mathcal{D}_v]^2, \tag{8}$$

*(iii) neighborhood second Zagreb index [27] is given by*

$$M_2^*(\mathcal{G}) = \sum_{uv \in E(\mathcal{G})} [\mathcal{D}_u \mathcal{D}_v], \tag{9}$$

*(iv) neighborhood forgotten topological index [27] is given by*

$$F^*(\mathcal{G}) = \sum_{uv \in V(\mathcal{G})} \mathcal{D}_v^3, \tag{10}$$

*(v) neighborhood modified version of forgotten topological index [27] is given by*

$$F_M^*(\mathcal{G}) = \sum_{uv \in E(\mathcal{G})} [\mathcal{D}_u^2 + \mathcal{D}_v^2], \tag{11}$$

*(vi) neighborhood harmonic index [28] is given by*

$$H^*(\mathcal{G}) = \sum_{uv \in E(\mathcal{G})} \frac{2}{\mathcal{D}_u + \mathcal{D}_v}, \tag{12}$$

*(vii)   neighborhood Randic index [28] is given by*

$$R^*(\mathcal{G}) = \sum_{uv \in E(\mathcal{G})} \frac{1}{\sqrt{\mathcal{D}_u \mathcal{D}_v}}, \tag{13}$$

*(viii)   neighborhood inverse Randic index [28] is given by*

$$RR^*(\mathcal{G}) = \sum_{uv \in E(\mathcal{G})} \sqrt{\mathcal{D}_u \mathcal{D}_v}. \tag{14}$$

The following are important inequalities used in establishing our results.

**Lemma 1** (Cauchy-Schwarz Inequality, [31]). *Let $x_i$ and $y_i$ be real numbers for all $1 \leq i \leq n$ then*

$$\left(\sum_{i=1}^n x_i y_i\right)^2 \leq \left(\sum_{i=1}^n x_i^2\right)\left(\sum_{i=1}^n y_i^2\right). \tag{15}$$

*Equality holds if and only if $x_i = k y_i$ for some constant $k$ for each $1 \leq i \leq n$.*

**Lemma 2** (Chebyshev's inequality, [31,32]). *Let $a_1 \leq a_2 \leq \ldots \leq a_n$ and $b_1 \leq b_2 \leq \ldots \leq b_n$ be real numbers then we have*

$$\sum_{i=1}^n a_i \sum_{i=1}^n b_i \leq n \sum_{i=1}^n a_i b_i. \tag{16}$$

*or*

$$\sum_{i=1}^n a_i b_i \geq \frac{1}{n} \sum_{i=1}^n a_i \sum_{i=1}^n b_i. \tag{17}$$

*Equality occurs if and only if $a_1 = a_2 = \ldots = a_n$ or $b_1 = b_2 = \ldots = b_n$.*

**Lemma 3** (Inequalities between Means, [31]). *Let $a, b \in \mathbb{R}^+$ and let*

$$QM = \sqrt{\frac{a^2 + b^2}{2}}, AM = \frac{a+b}{2}, GM = \sqrt{ab}, HM = \frac{2}{\frac{1}{a} + \frac{1}{b}}, \tag{18}$$

*then $QM \geq AM \geq GM \geq HM$, equality occurs if and only if $a = b$ where QM = quadratic mean, AM = arithmetic mean, GM = geometric mean, and HM = harmonic mean.*

**Lemma 4** ([31]). *Let $a_1, a_2, \ldots, a_n$ and $b_1, b_2, \ldots, b_n$ be two sequences of non-negative real numbers and $c_i > 0$, $i = 1, 2, \ldots, n$ such that $\frac{a_1}{c_1} \geq \frac{a_2}{c_2} \geq \ldots \geq \frac{a_n}{c_n}$ and $\frac{b_1}{c_1} \geq \frac{b_2}{c_2} \geq \ldots \geq \frac{b_n}{c_n}$ then*

$$\sum_{i=1}^n \frac{a_i b_i}{c_i} \geq \frac{\sum_{i=1}^n a_i \sum_{i=1}^n b_i}{\sum_{i=1}^n c_i}. \tag{19}$$

**Lemma 5** (Diaz-Metcalf inequality, [24]). *Let $a_i$ and $b_i$ be two sequences of real numbers with $a_i \neq 0 (i = 1, 2, \ldots, n)$ and such that $p a_i \leq b_i \leq P a_i$ then*

$$\sum_{i=1}^n b_i^2 + p P \sum_{i=1}^n a_i^2 \leq (P + p) \sum_{i=1}^n a_i b_i. \tag{20}$$

*Equality holds if and only if either $b_i = p a_i$ or $b_i = P a_i$ for every $i = 1, 2, \ldots, n$.*

## 3. Main Results

We begin this section by introducing neighborhood versions of two degree based topological indices, namely the GA index and ABC index. We establish their bounds for some graphs.

**Definition 2.** *Let $\mathcal{G}$ be a graph and let $u, v \in V(\mathcal{G})$ and $\mathcal{D}_u, \mathcal{D}_v$ be the neighbor degree sums of vertices u and v, respectively; then, the neighborhood GA index is given by*

$$NGA(\mathcal{G}) = \sum_{u,v \in E(\mathcal{G})} \frac{2\sqrt{\mathcal{D}_u \mathcal{D}_v}}{\mathcal{D}_u + \mathcal{D}_v}, \tag{21}$$

*where $\mathcal{D}_u = \sum_{v \in N(v)} d(v)$.*

**Definition 3.** *Let $\mathcal{G}$ be a graph and let $u, v \in V(\mathcal{G})$ and $\mathcal{D}_u, \mathcal{D}_v$ be the neighbor degree sums of vertices u and v, respectively; then, the neighborhood ABC index is given by*

$$NABC(\mathcal{G}) = \sum_{uv \in E(\mathcal{G})} \sqrt{\frac{\mathcal{D}_u + \mathcal{D}_v - 2}{\mathcal{D}_u \mathcal{D}_v}}, \tag{22}$$

*where $\mathcal{D}_u = \sum_{v \in N(v)} d(v)$.*

Using the above definitions, we compute the bounds of $NGA(\mathcal{G})$ and $NABC(\mathcal{G})$ for some classical graphs such as $R_n$, $K_n$, $C_n$, $S_n$, irregular graphs and pendant graphs. The following remarks will be important in establishing our result.

**Remark 1.** *We denote $\max(\mathcal{D}_u) = \Delta$ and $\min(\mathcal{D}_u) = \delta$. If $\mathcal{G}$ is a complete, regular, cycle or star graph then $\max(\mathcal{D}_u) = \min(\mathcal{D}_u)$ i.e., $\Delta = \delta$.*

**Remark 2.** *If $\mathcal{G}$ is a complete, regular or cycle graph then $\mathcal{D}_u = \mathcal{D}_v = d_v^2$.*

*3.1. Bounds for Neighborhood Geometric–Arithmetic (NGA) Index of Graphs*

**Theorem 1.** *Let $\mathcal{G}$ be a complete, regular, cycle or star graph with m-edges then*

$$NGA(\mathcal{G}) = m. \tag{23}$$

**Proof.** From Remark 2, we have

$$NGA(\mathcal{G}) = \sum_{uv \in E(\mathcal{G})} \frac{2\sqrt{\mathcal{D}_u \mathcal{D}_v}}{\mathcal{D}_u + \mathcal{D}_v} = \sum_{uv \in E(\mathcal{G})} \frac{2d_v^2}{2d_v^2} = \sum_{uv \in E(\mathcal{G})} 1. \tag{24}$$

Since $\mathcal{G}$ is a $K_n, R_n, C_n$ or $S_n$ graph then the neighborhood degree sum of a pair $u, v \in E(\mathcal{G})$ is 1. Therefore, if we take the summation of the neighborhood degree for all $uv \in E(\mathcal{G})$ we obtain

$$NGA(\mathcal{G}) = m. \tag{25}$$

$\square$

**Theorem 2.** *Let $\mathcal{G}$ be a star graph then*

$$NGA(\mathcal{G}) \geq \frac{2\sqrt{n-1}}{n}. \tag{26}$$

**Proof.** Let $\mathcal{G}$ be a star graph and $1 \leq \mathcal{D}_u \leq n - 1$ holds for $u \in V(\mathcal{G})$. Since

$$NGA(\mathcal{G}) = \sum_{uv \in E(\mathcal{G})} \frac{2\sqrt{\mathcal{D}_u \mathcal{D}_v}}{\mathcal{D}_u + \mathcal{D}_v}, \tag{27}$$

then it follows from Lemma 3 that

$$\sum_{uv \in E(\mathcal{G})} 2\sqrt{\mathcal{D}_u \mathcal{D}_v} \leq \sum_{uv \in E(\mathcal{G})} (\mathcal{D}_u + \mathcal{D}_v),$$

which implies that

$$\sum_{uv \in E(\mathcal{G})} \frac{\mathcal{D}_u + \mathcal{D}_v}{2} \geq \sum_{uv \in E(\mathcal{G})} \sqrt{\mathcal{D}_u \mathcal{D}_v}. \tag{28}$$

Let $a = \mathcal{D}_u = 1$ and $b = \mathcal{D}_v = n - 1$, substituting these in (28) we obtain

$$\sum_{uv \in E(\mathcal{G})} \frac{1 + n - 1}{2} \geq \sum_{uv \in E(\mathcal{G})} \sqrt{1.(n-1)},$$

which implies

$$m \geq \frac{2\sqrt{n-1}}{n}, \tag{29}$$

and therefore from Theorem (23) we obtain

$$NGA(\mathcal{G}) \geq \frac{2\sqrt{n-1}}{n}. \tag{30}$$

$\square$

**Theorem 3.** *Let $\mathcal{G}$ be a complete, regular or cycle graph with m-edges. Then,*

$$NGA(\mathcal{G}) \geq \frac{2m^2}{R^*(\mathcal{G})M_1^*(\mathcal{G})}. \tag{31}$$

**Proof.** From Definition 2, if $a = \sqrt{\mathcal{D}_u \mathcal{D}_v}$ and $b = \frac{1}{2}(\mathcal{D}_u + \mathcal{D}_v)$ then

$$\sum_{uv \in E(\mathcal{G})} 1 = \sum_{uv \in E(\mathcal{G})} \left( \sqrt{\frac{a}{b}} \times \frac{1}{\sqrt{\frac{a}{b}}} \right). \tag{32}$$

Squaring and applying Lemma 1 to (32)

$$\left( \sum_{uv \in E(\mathcal{G})} 1 \right)^2 \leq \left( \sum_{uv \in E(\mathcal{G})} \sqrt{\frac{a}{b}} \right)^2 \times \left( \sum_{uv \in E(\mathcal{G})} \frac{1}{\sqrt{\frac{a}{b}}} \right)^2, \tag{33}$$

which becomes

$$m^2 \leq \sum_{uv \in E(\mathcal{G})} \frac{\sqrt{\mathcal{D}_u \mathcal{D}_v}}{\frac{1}{2}(\mathcal{D}_u + \mathcal{D}_v)} \times \sum_{uv \in E(\mathcal{G})} \frac{\frac{1}{2}(\mathcal{D}_u + \mathcal{D}_v)}{\sqrt{\mathcal{D}_u \mathcal{D}_v}} \leq NGA(\mathcal{G}) \times \sum_{uv \in E(\mathcal{G})} \frac{1}{2}.\frac{\mathcal{D}_u + \mathcal{D}_v}{\sqrt{\mathcal{D}_u \mathcal{D}_v}}, \tag{34}$$

which implies

$$2m^2 \leq NGA(\mathcal{G}) \times \sum_{uv \in E(\mathcal{G})} \frac{1}{\sqrt{\mathcal{D}_u \mathcal{D}_v}}.(\mathcal{D}_u + \mathcal{D}_v). \tag{35}$$

From (13) and (7), we have

$$2m^2 \leq NGA(\mathcal{G}) \times R^*(\mathcal{G})M_1^*(\mathcal{G}). \tag{36}$$

Therefore,

$$NGA(\mathcal{G}) \geq \frac{2m^2}{R^*(\mathcal{G})M_1^*(\mathcal{G})}. \tag{37}$$

This completes the proof. $\square$

**Theorem 4.** *Let $\mathcal{G}$ be a pendant graph then*

$$NGA(\mathcal{G}) \leq \frac{\sqrt{2\Delta\delta}H^*(\mathcal{G})RR^*(\mathcal{G})}{\sqrt{\Delta^2 + \delta^2}}. \tag{38}$$

**Proof.** Squaring (21) and applying Lemma 1, we have

$$[NGA(\mathcal{G})]^2 = \left(\sum_{uv\in E(\mathcal{G})} \frac{2\sqrt{\mathcal{D}_u\mathcal{D}_v}}{\mathcal{D}_u + \mathcal{D}_v}\right)^2 \leq \frac{\sum\limits_{uv\in E(\mathcal{G})} (\mathcal{D}_u\mathcal{D}_v) \sum\limits_{uv\in E(\mathcal{G})} \frac{4}{(\mathcal{D}_u+\mathcal{D}_v)^2}}{\frac{1}{2}\left(\frac{\Delta}{\delta} + \frac{\delta}{\Delta}\right)}, \tag{39}$$

which implies that

$$NGA(\mathcal{G}) \leq \sqrt{\sum_{uv\in E(\mathcal{G})} \mathcal{D}_u\mathcal{D}_v . \sum_{uv\in E(\mathcal{G})} \frac{4}{(\mathcal{D}_u + \mathcal{D}_v)^2} \times \frac{2\Delta\delta}{\Delta^2 + \delta^2}}$$

$$= \sum_{uv\in E(\mathcal{G})} \frac{2}{\mathcal{D}_u + \mathcal{D}_v} . \sum_{uv\in E(\mathcal{G})} \sqrt{\mathcal{D}_u\mathcal{D}_v} \times \sqrt{\frac{2\Delta\delta}{\Delta^2 + \delta^2}}.$$

Hence,

$$NGA(\mathcal{G}) \leq \frac{\sqrt{2\Delta\delta}H^*(\mathcal{G})RR^*(\mathcal{G})}{\sqrt{\Delta^2 + \delta^2}}. \tag{40}$$

□

**Theorem 5.** *Let $\mathcal{G}$ be a any graph*

$$NGA(\mathcal{G}) \leq H^*(\mathcal{G})\sqrt{M_2^*(\mathcal{G})}. \tag{41}$$

**Proof.** From Definition 2, let $a = \frac{2}{\mathcal{D}_u + \mathcal{D}_v}$ and $b = \sqrt{\mathcal{D}_u\mathcal{D}_v}$. Squaring (21) and applying Lemma 1, we have

$$[NGA(\mathcal{G})]^2 = \left(\sum_{uv\in E(\mathcal{G})} \frac{2}{\mathcal{D}_u + \mathcal{D}_v} . \sqrt{\mathcal{D}_u\mathcal{D}_v}\right)^2 \leq \left(\sum_{uv\in E(\mathcal{G})} \frac{2}{\mathcal{D}_u + \mathcal{D}_v}\right)^2 \left(\sum_{uv\in E(\mathcal{G})} \sqrt{\mathcal{D}_u\mathcal{D}_v}\right)^2. \tag{42}$$

which implies

$$[NGA(\mathcal{G})]^2 \leq \left(\sum_{uv\in E(\mathcal{G})} \frac{2}{\mathcal{D}_u + \mathcal{D}_v}\right)^2 \left(\sum_{uv\in E(\mathcal{G})} \mathcal{D}_u\mathcal{D}_v\right).$$

$$[NGA(\mathcal{G})]^2 \leq [H^*(\mathcal{G})]^2 . M_2^*(\mathcal{G}).$$

Taking square root of both sides yields

$$NGA(\mathcal{G}) \leq H^*(\mathcal{G})\sqrt{M_2^*(\mathcal{G})}. \tag{43}$$

□

**Theorem 6.** *Let $\mathcal{G}$ be any graph then*

$$NGA(\mathcal{G}) \geq \frac{H^*(\mathcal{G})^2 + \Delta\delta M_2^*(\mathcal{G})}{\Delta + \delta}. \tag{44}$$

**Proof.** From Definition 2, let $a = \sqrt{\mathcal{D}_u \mathcal{D}_v}$, $b = \frac{2}{\mathcal{D}_u + \mathcal{D}_v}$, $p = \delta$, $P = \Delta$. Applying Lemma 5 to (21), we have

$$\sum_{uv \in E(\mathcal{G})} \left( \frac{2}{\mathcal{D}_u + \mathcal{D}_v} \right)^2 + \delta\Delta \sum_{uv \in E(\mathcal{G})} \left( \sqrt{\mathcal{D}_u \mathcal{D}_v} \right)^2 \leq (\Delta + \delta) \sum_{uv \in E(\mathcal{G})} \sqrt{\mathcal{D}_u \mathcal{D}_v} \cdot \frac{2}{\mathcal{D}_u + \mathcal{D}_v}. \quad (45)$$

From (12) and (9), we have

$$H^*(\mathcal{G})^2 + \Delta\delta M_2^*(\mathcal{G}) \leq (\Delta + \delta) \sum_{uv \in E(\mathcal{G})} \frac{2\sqrt{\mathcal{D}_u \mathcal{D}_v}}{\mathcal{D}_u + \mathcal{D}_v}.$$

Hence,

$$\frac{H^*(\mathcal{G})^2 + \Delta\delta M_2^*(\mathcal{G})}{\Delta + \delta} \leq NGA(\mathcal{G}). \quad (46)$$

$\square$

**Theorem 7.** *Let $\mathcal{G}$ be a star graph with m-edges then*

$$NGA(\mathcal{G}) \geq \frac{RR^*(\mathcal{G})H^*(\mathcal{G})}{m}. \quad (47)$$

**Proof.** From Definition 2, let $a = \sqrt{\mathcal{D}_u \mathcal{D}_v}$ and $b = \frac{2}{\mathcal{D}_u + \mathcal{D}_v}$. Applying Lemma 2 to (21), we have

$$\left( \sum_{uv \in E(\mathcal{G})} \sqrt{\mathcal{D}_u \mathcal{D}_v} \right) \cdot \left( \sum_{uv \in E(\mathcal{G})} \frac{2}{\mathcal{D}_u + \mathcal{D}_v} \right) \leq m \left( \sum_{uv \in E(\mathcal{G})} \frac{2}{\mathcal{D}_u + \mathcal{D}_v} \cdot \sqrt{\mathcal{D}_u \mathcal{D}_v} \right). \quad (48)$$

$$\frac{\left( \sum_{uv \in E(\mathcal{G})} \sqrt{\mathcal{D}_u \mathcal{D}_v} \right) \left( \sum_{uv \in E(\mathcal{G})} \frac{2}{\mathcal{D}_u + \mathcal{D}_v} \right)}{m} \leq \left( \sum_{uv \in E(\mathcal{G})} \frac{2}{\mathcal{D}_u + \mathcal{D}_v} \cdot \sqrt{\mathcal{D}_u \mathcal{D}_v} \right). \quad (49)$$

It implies from (49) and by (14) and (12), we get

$$\frac{RR^*(\mathcal{G})H^*(\mathcal{G})}{m} \leq \left( \sum_{uv \in E(\mathcal{G})} \frac{2\sqrt{\mathcal{D}_u \mathcal{D}_v}}{\mathcal{D}_u + \mathcal{D}_v} \right). \quad (50)$$

Therefore

$$\frac{RR^*(\mathcal{G})H^*(\mathcal{G})}{m} \leq NGA(\mathcal{G}). \quad (51)$$

$\square$

*3.2. Bounds for Neighborhood Atom Bond Connectivity $NABC$ Index of Graphs*

The following remark is an analogue of the properties of the classical $ABC$ index studied by Das et al. [24]. The properties also follows for neighborhood version of $ABC$ index.

**Remark 3.** *If $\mathcal{G}$ is a complete, regular, cycle or star graph then*

$$\mathcal{D}_u \mathcal{D}_v (\mathcal{D}_u + \mathcal{D}_v - 2) = 2\Delta^2 (\Delta - 1) \quad (52)$$

**Theorem 8.** *Let $\mathcal{G}$ be any graph then*

$$\mathcal{D}_u + \mathcal{D}_v - 2 \geq \frac{2m\Delta^2(\Delta - 1)}{M_2^*(\mathcal{G})}. \tag{53}$$

**Proof.** From Remark 3, the following

$$\mathcal{D}_u + \mathcal{D}_v - 2 = \frac{2\Delta^2(\Delta - 1)}{\mathcal{D}_u \mathcal{D}_v} \tag{54}$$

holds. Taking the summation of (54) we have

$$\sum_{uv \in E(\mathcal{G})} \mathcal{D}_u + \mathcal{D}_v - 2 = \sum_{uv \in E(\mathcal{G})} \frac{2\Delta^2(\Delta - 1)}{\mathcal{D}_u \mathcal{D}_v}. \tag{55}$$

Using Lemma 4 on (55), let $a = 2\Delta^2(\Delta - 1)$ and let $b = 1$ and $c = \mathcal{D}_u \mathcal{D}_v$; then, we have

$$\sum_{uv \in E(\mathcal{G})} \frac{2\Delta^2(\Delta - 1).1}{\mathcal{D}_u \mathcal{D}_v} \geq \frac{2 \sum\limits_{uv \in E(\mathcal{G})} \Delta^2(\Delta - 1). \sum\limits_{uv \in E(\mathcal{G})} 1}{\sum\limits_{uv \in E(\mathcal{G})} \mathcal{D}_u \mathcal{D}_v} \geq \frac{2m\Delta^2(\Delta - 1)}{M_2^*(\mathcal{G})}. \tag{56}$$

We obtain from (54) and (56) that

$$\sum_{uv \in E(\mathcal{G})} \mathcal{D}_u + \mathcal{D}_v - 2 \geq \frac{2m\Delta^2(\Delta - 1)}{M_2^*(\mathcal{G})}. \tag{57}$$

□

**Theorem 9.** *Let $\mathcal{G}$ be a star graph then,*

$$NABC(\mathcal{G}) = \sqrt{2(m - 1)}. \tag{58}$$

**Proof.** From Definition 3, let $\mathcal{D}_u = \mathcal{D}_v = m$ and

$$NABC(\mathcal{G}) = \sum_{uv \in E(\mathcal{G})} \sqrt{\frac{\mathcal{D}_u + \mathcal{D}_v - 2}{\mathcal{D}_u \mathcal{D}_v}} = \frac{\sqrt{2(m - 1)}}{m}. \tag{59}$$

Since $\mathcal{G}$ is a star graph then the neighborhood degree sum of a pair $u, v \in E(\mathcal{G})$ is $\frac{\sqrt{2(m-1)}}{m}$. Therefore, if we take the summation of the neighborhood degree for all $uv \in E(\mathcal{G})$ we obtain

$$\sum_{uv \in E(\mathcal{G})} \frac{\sqrt{2(m - 1)}}{m} \times m = \sqrt{2(m - 1)}. \tag{60}$$

□

**Theorem 10.** *Let $\mathcal{G}$ be any graph then*

$$NABC(\mathcal{G}) \leq \Delta R^*(\mathcal{G})\sqrt{\frac{2m(\Delta - 1)}{M_2^*(\mathcal{G})}}. \tag{61}$$

*Equality holds if and only if $\mathcal{G}$ is a regular, complete, or star graph.*

**Proof.** Squaring (22) and applying Lemma 1, let $a = \sqrt{\mathcal{D}_u + \mathcal{D}_v - 2}$ and $b = \frac{1}{\sqrt{\mathcal{D}_u \mathcal{D}_v}}$ then

$$\left( \sum_{uv \in E(\mathcal{G})} \sqrt{\frac{\mathcal{D}_u + \mathcal{D}_v - 2}{\mathcal{D}_u \mathcal{D}_v}} \right)^2 \leq \left( \sum_{uv \in E(\mathcal{G})} \sqrt{\mathcal{D}_u + \mathcal{D}_v - 2} \right)^2 \left( \sum_{uv \in E(\mathcal{G})} \frac{1}{\sqrt{\mathcal{D}_u \mathcal{D}_v}} \right)^2 \tag{62}$$

$$\leq \sum_{uv \in E(\mathcal{G})} \mathcal{D}_u + \mathcal{D}_v - 2 \sum_{uv \in E(\mathcal{G})} \left( \frac{1}{\sqrt{\mathcal{D}_u \mathcal{D}_v}} \right)^2. \tag{63}$$

From (53) and (13), (63) becomes

$$\left( \sum_{uv \in E(\mathcal{G})} \sqrt{\frac{\mathcal{D}_u + \mathcal{D}_v - 2}{\mathcal{D}_u \mathcal{D}_v}} \right)^2 \leq \frac{2m\Delta^2(\Delta - 1)}{M_2^*(\mathcal{G})} \cdot [R^*(\mathcal{G})]^2.$$

Taking square root of both sides,

$$\sum_{uv \in E(\mathcal{G})} \sqrt{\frac{\mathcal{D}_u + \mathcal{D}_v - 2}{\mathcal{D}_u \mathcal{D}_v}} \leq \Delta R^*(\mathcal{G}) \sqrt{\frac{2m(\Delta - 1)}{M_2^*(\mathcal{G})}}. \tag{64}$$

Hence,

$$NABC(\mathcal{G}) \leq \Delta R^*(\mathcal{G}) \sqrt{\frac{2m(\Delta - 1)}{M_2^*(\mathcal{G})}}. \tag{65}$$

□

**Theorem 11.** *Let $\mathcal{G}$ be any graph and $m$ the number of edges in $\mathcal{G}$. Then,*

$$NABC(\mathcal{G}) \geq \frac{\Delta R^*(\mathcal{G})}{m} \sqrt{\frac{2m(\Delta - 1)}{M_2^*(\mathcal{G})}}. \tag{66}$$

**Proof.** From Lemma 2, let $a = \sqrt{\mathcal{D}_u + \mathcal{D}_v - 2}$ and $b = \frac{1}{\sqrt{\mathcal{D}_u \mathcal{D}_v}}$

$$\sum_{uv \in E(\mathcal{G})} \left( \sqrt{\mathcal{D}_u + \mathcal{D}_v - 2} \cdot \frac{1}{\sqrt{\mathcal{D}_u \mathcal{D}_v}} \right) \geq \left( \frac{1}{m} \sum_{uv \in E(\mathcal{G})} \sqrt{\mathcal{D}_u + \mathcal{D}_v - 2} \sum_{uv \in E(\mathcal{G})} \frac{1}{\sqrt{\mathcal{D}_u \mathcal{D}_v}} \right). \tag{67}$$

From (53) and (13), we have

$$\sum_{uv \in E(\mathcal{G})} \sqrt{\frac{\mathcal{D}_u + \mathcal{D}_v - 2}{\mathcal{D}_u \mathcal{D}_v}} \geq \frac{1}{m} \sqrt{\frac{2m\Delta^2(\Delta - 1)}{M_2^*(\mathcal{G})}} \cdot R^*(\mathcal{G}).$$

Hence

$$NABC(\mathcal{G}) \geq \frac{\Delta R^*(\mathcal{G})}{m} \sqrt{\frac{2m(\Delta - 1)}{M_2^*(\mathcal{G})}}. \tag{68}$$

□

We conclude this section with the following consequence from the classical and neighborhood versions of *ABC* and *GA* indices.

**Corollary 1.** *Let $GA$, $ABC$, $NGA$ and $NABC$ be geometric–arithmetic index, atom bond connectivity index, neighborhood geometric–arithmetic index, and neighborhood atom bond connectivity index; then, the following inequalities hold:*

*(i)　If $\mathcal{G}$ is a star graph, then $NGA(\mathcal{G}) \geq GA(\mathcal{G})$.*
*(ii)　If $\mathcal{G}$ is a regular, cycle, or complete graph, then $NGA(\mathcal{G}) = GA(\mathcal{G})$.*
*(iii)　If $\mathcal{G}$ is a pendant graph, then $NGA(\mathcal{G}) \leq GA(\mathcal{G})$.*

(iv)   If $\mathcal{G}$ is an irregular graph, then $NGA(\mathcal{G}) \leq GA(\mathcal{G})$.
(v)    If $\mathcal{G}$ is any graph, then $NABC(\mathcal{G}) \geq ABC(\mathcal{G})$.

## 4. Conclusions

In this article, we have introduced and studied the neighborhood geometric–arithmetic index and neighborhood atom bond connectivity index for some known graphs. We have established the lower and upper bounds of $NGA(\mathcal{G})$ and $NABC(\mathcal{G})$ for some known graphs. In our future studies, we are going to demonstrate the priority and usage of our new neighborhood topological indices against other types of topological indices in QSPR/QSAR analysis of drugs. Unlike the normal SPSS adopted for QSPR/QSAR, we plan to adopt the support vector machine algorithm of machine learning.

**Author Contributions:** Conceptualization of the article was given by M.S.A., K.O.A., and M.A.; methodology by M.S.A. and K.O.A.; validation by K.O.A. and M.A.; formal analysis, investigation, data curation, and writing—original draft preparation by M.S.A., K.O.A., and M.A.; resources by K.O.A. and M.A.; writing—review and editing by M.S.A., K.O.A., and M.A.; visualization by K.O.A. and M.A.; and project administration by K.O.A. All authors have read and agreed to the published version of the manuscript.

**Funding:** The APC was funded by the Department of Mathematics and Applied Mathematics, Sefako Makgatho Health Sciences, Pretoria, South Africa.

**Data Availability Statement:** Not applicable.

**Conflicts of Interest:** The authors declare no conflict of interest.

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
