# Peer review of "Neighborhood Versions of Geometric–Arithmetic and Atom Bond Connectivity Indices of Some Popular Graphs and Their Properties"

_axioms, doi:10.3390/axioms11090487_

Round 1

Reviewer 1 Report

This is a systematic, perhaps even meticulous, study adding some new "neighborhood" graph-theoretic indices for molecular systems. While I do not find the analysis to be exceptional, it is meritorious, and given the increasing use of graph-neural-networks in a molecular context, may be extremely valuable as features in modern-data QSPR/QSAR using machine-learning. 

I wish the authors had provided even a small, cursory, example showing good performance for the new indicators, but the paper is more of an idea+proof paper, and is meritorious by that criterion. In my opinion, it can be published as is.

Reviewer 2 Report

The article proposes alternative (neighborhood versions) indices to the two classical topological indices in the graph. It is an extension of geometric-arithmetic and neighborhood atom bond connectivity indices. Indexes are defined for all basic graph types (complete graph, regular graph, cycle graph, star graph, pendant graph and irregular graph). Formulas and proofs of correctness are defined in all cases given. The research results can be applied, for example, in the field of chemistry when examining the structures of molecules. The article is written methodically correctly and comprehensibly. I only have a comment about the "Conclusion" section, which is too brief. I recommend to include in this section a comparison with other similar topological indices in terms of priority and usage and suggest further research in this area.

Reviewer 3 Report

In this manuscript the authors introduce two new indeces related to graphs and present some lower and upper bounds for some particular types of graphs. I really liked the introductory part, where the state-of-the-art is very nicely presented: it is not just a historic presentation of the studies done on this argument, but also their motivation is mentioned. The reader understands why some indices were introduced and studied. Unfortunately, this motivation misses in the current study. We don't know why it is important/useful/interesting to define other two new indices. Their role is not mentioned at all and the provided bounds are sometimes too complex. Without this explanation the study has a very poor significance and the overall merit is pretty low.

The authors must provide some applications of the new indices in order to give a significance/motivation of their study.

Based on this, I reject the paper and encourage the resubmision of an improved version.

Round 2

Reviewer 3 Report

I am sorry, but the answers of the authors do not convince me to change my idea. From this manuscript, I don't see any clear motivation of this study, no applications are provided. The role and importance of this study remain unclear. Based on this, I propose the rejection.